# Molecular Simulation of Interactions between High-Molecular-Polymer Flocculation Gel for Oil-Based Drilling Fluid and Clay Minerals

**DOI:** 10.3390/gels8070442

**Published:** 2022-07-15

**Authors:** Zhijun He, Jintang Wang, Bo Liao, Yujing Bai, Zihua Shao, Xianbin Huang, Qi Wang, Yiyao Li

**Affiliations:** School of Petroleum Engineering, China University of Petroleum (East China), Qingdao 266580, China; 1902010130@s.upc.edu.cn (Z.H.); liaob@outlook.com (B.L.); 18435681348@163.com (Y.B.); 1702010801@s.upc.edu.cn (Z.S.); 20170092@upc.edu.cn (X.H.); 15610571573@163.com (Q.W.); 1902010117@s.upc.edu.cn (Y.L.)

**Keywords:** drilling fluid, clay minerals, flocculation gel, molecular simulation

## Abstract

China has abundant shale gas resources with great potential, which may serve as a significant support for the development of a “low-carbon economy”. Domestic shale gas resources are buried deeply and difficult to exploit due to some prevalent issues, such as long horizontal sections, severe development of reservoir fractures, strong sensitivity to water, borehole instability, etc. Compared to water-based drilling fluids, oil-based drilling fluid exhibits better inhibition and good lubricity and is thus broadly used in shale gas drilling, but it is confronted with the challenge of removing the harmful solid phase. Selective chemical flocculation is one of the most effective methods of removing the harmful solid phase in oil-based drilling fluid. In this study, interactions between the flocculation gel for oil-based drilling fluid and clay minerals were investigated by molecular simulation, which revealed the molecular-scale selectivity of the flocculation gel for rock cuttings with negative charges. Calculations showed that the flocculation gel is highly effective for the flocculation of negatively charged cuttings, but it is ineffective for flocculating neutral cuttings. The flocculation gel is not very effective for cuttings with high hydrophilicity, and it is totally ineffective for flocculating cuttings with poor hydrophilicity. Within a limited concentration range, the flocculation effect can be enhanced by increasing the flocculation gel concentration. The performance of the flocculation gel declined at elevated temperatures.

## 1. Introduction

Shale gas is a high-quality, efficient, and clean low-carbon energy resource serving as a significant support for developing a “low-carbon economy”. China has abundant shale gas resources with great potential [1]. The large-scale exploitation of shale gas and the acceleration of the development of the shale gas industry have become the focuses of oil-and-gas exploration and development in China, which has great strategic significance for ensuring the energy supply, achieving self-sufficiency, improving the ecological environment, and constructing a clean, low-carbon, safe, and efficient energy system in China [2].

Shale gas resources in China are deeply buried and difficult to exploit [3]. Some of the prevalent issues hindering shale gas exploitation include the long horizontal sections, the severe development of reservoir fractures, strong water-sensitivity, borehole instability, etc. [4] Compared with water-based drilling fluid, oil-based drilling fluid exhibits better inhibition and good lubricity. Owing to these advantages, oil-based drilling fluid has become the preferred drilling-fluid system for the horizontal drilling of sections of shale gas [5]. The application of oil-based drilling fluid is confronted with three major problems in the field which have not yet been fully addressed. First, the harmful solid phase in the drilling fluid system is hard to remove. Second, the drilling fluid is not highly effective when reused. Third, drilling cuttings removed by solid control equipment (e.g., vibrating screens, desanders, and cleaners) still have a high content of nano/micron-sized harmful solid phase and high-viscosity drilling fluid, even after the secondary removal of the coarse, harmful solid phase by high-speed centrifuging. Hence, the drilling fluid cannot be reused and must be discarded, which not only causes a serious waste of drilling fluid, but it also has a high disposal cost due to the hazardous nature of the waste, as well as the risks of environmental pollution [6,7].

Selective chemical flocculation is an effective method of removing the harmful solid phase. Selective flocculation involves adding a polymer flocculation gel to the drilling fluid, where the flocculation gel will selectively flocculate the harmful solid phase, but it has no flocculation effects on barite and bentonite. The harmful solid phase can then be eliminated by incorporating solid-control equipment [8,9]. Experimental approaches are often adopted to explore flocculation mechanisms. However, the mechanism of the interaction between the flocculation gel and clay minerals cannot be deeply analyzed experimentally. Exploiting molecular simulation methods in developing flocculation gel can provide guidance for understanding the mechanism of selective flocculation on the micro-scale [10,11,12,13,14]. The main functional mechanism of flocculation gel is considered to be the adsorption-bridging effect, i.e., chain-structure flocculation gel with a high molecular weight is adsorbed on the surface of multiple, suspended particles simultaneously so as to form a bridging structure, which results in flocculation by pulling multiple particles together [12,15]. Therefore, the agent molecule used as the flocculation gel should have suitable groups and molecular structures for strong adsorption on the surface of suspended particles to enable the sufficiently strong adsorption on the surface of the suspended particles. Meanwhile, the flocculation gel molecules should be chain molecules with a small number of branches or cross-linked structures, and should have a relatively large molecular weight, generally around 10^6^, to form bridging structures between different suspended particles [16].

From the molecular perspective, a segment of flocculation gel with a certain length was adopted to simulate adsorption on the surface of suspended particles and investigate the adsorption behaviors of flocculation gel molecules on the surface of suspended rock cuttings, as well as to calculate the corresponding interaction energy.

In this study, the generation of nano/micron-sized drilling cuttings during the drilling process and selective flocculation by a flocculation gel for oil-based drilling fluid was investigated via molecular simulation. By calculating the density distribution, 3D molecular conformation, diffusion coefficient, and interaction energy of the flocculated system, the mechanism of the selective flocculation of the cuttings by the flocculation gel in the oil-based drilling fluid was analyzed. The effectiveness of the flocculation gel for the oil-based drilling fluid was investigated as a function of the concentration and temperature. The data can provide theoretical support for developing high-efficiency flocculation gel.

## 2. Results and Discussion

### 2.1. Simulation of Selective Flocculation of Rock Cuttings

The density distribution of the flocculation gel in the system was statistically analyzed (Figure 1). According to the density-distribution diagram, in the case of montmorillonite, the flocculation gel density was maximal 0.51 nm away from the surface, while the corresponding distance was 2.02 nm and 2.89 nm for the hydroxyl surface and silica surface of kaolinite. For the three types of surfaces, the flocculation gel density was 0.085, 0.045, and 0.038 g/cm^3^, respectively. Therefore, it can be deduced that for montmorillonite, the flocculation gel distribution is closer to the surface of the solid phase of the rock cuttings, and the flocculation gel concentration is also higher.

The 3D configuration schematics of the particle distribution in the system at 4.0 ns are shown in Figure 2. The overall characteristics of the kaolinite surface are hydrophilic, with multiple, relatively strong, hydrogen bonds and surface hydroxyl groups on the octahedral surface, and the contact angle of the surface is equal to zero, so the surface has strong hydrophilicity, while the four-sided surface has strong hydrophobicity. There are numerous siloxy groups on the body, and the water molecules on the surface are held together, maintaining the deformed shape of the droplet, indicating that the tetrahedral kaolinite surface is hydrophobic [17,18]. According to the 3D configuration schematics, the interaction between the flocculation gel and electrically neutral surfaces with weak hydrophilicity was relatively weak, but the flocculation gel had good extensibility. In the case of electriferous surfaces, there was a mutual attraction between the flocculation gel and the electriferous surface. The flocculation gel interacted with a large number of adsorption sites on the surface, thereby tending to tile or wrap the cuttings. As cationic groups containing N atoms were the closest to the surface, it can be inferred that the coulomb adsorption between the electriferous groups of the flocculation gel and electrically negative montmorillonite particles was the major force driving the interactions.

The diffusion coefficient of the flocculation gel after flocculating the rock cuttings is shown in Figure 3. The results indicate that the diffusion coefficient of the flocculation gel was the lowest after flocculation in the system of montmorillonite particles, but it was the largest in the system of kaolinite particles with silica surfaces [19]. That is, the diffusion ability of the flocculation gel declined after flocculating the rock cuttings. Therefore, it can be further deduced that the migration ability of the flocculation gel in the system declined during the flocculation process. With more electriferous components in the system, flocculation would be more effective.

The correlation between the flocculation gel and clay minerals in different systems is demonstrated in Figure 4, where a larger interaction energy indicates a stronger interaction. The interaction energy between the flocculation gel and kaolinite particles with hydroxyl and silica surfaces was determined as −0.1883 and −0.0001 kcal/mol, respectively, while the interaction energy between the flocculation gel and montmorillonite particles was calculated as −260.5685 kcal/mol. Therefore, adsorption of the flocculation gel on the surface of montmorillonite is much more significant than that on kaolinite. Thus, the flocculation gel had little effect on non-hydrophilic rock cuttings without charges. In summary, the selective adsorption of the flocculation gel on the cuttings was mainly related to coulomb adsorption. The strength of the adsorption of the flocculation gel on different surfaces followed this order: electriferous surface > strongly hydrophilic surface > weakly hydrophilic surface. Montmorillonite had more adsorption sites for the flocculation gel.

### 2.2. Effect of Flocculation Gel Concentrations on Flocculation Performance

To investigate the effect of the flocculation gel concentration on the flocculation efficiency, different molecular models of the flocculation gel were built, as shown in Figure 5. The number of flocculation gel molecules in the systems was 1, 10, and 40, respectively, and the density distribution in the systems with different flocculation gel concentrations was statistically calculated, as illustrated in Figure 6. In the system, the region 1.0 nm away from the surface was considered as the adsorption phase, while the region with a distance of more than 1.0 nm was considered as the bulk phase. According to the figure, the concentration of the flocculation gel adsorption phase near the cuttings increased as the molecular concentration of the flocculation gel increased. When 40 flocculation gel molecules were added, the total concentration of flocculation gel in the bulk phase also increased, which indicates that the concentration of flocculation gel in the adsorption phase reached the saturation point.

Figure 7 shows the variation of the interaction energy in the flocculation gel systems with different concentrations. The results indicated that the total interaction between the flocculation gel and cuttings was enhanced with an increase in the number of flocculation gel molecules, which could enhance the effectiveness of flocculation. However, the average interaction between the flocculation gel molecules and cuttings decreased, i.e., the flocculation performance of a single flocculation gel molecule declined with an increase in the number of flocculation gel molecules. Combining the density distribution of the flocculation gel on the surface of the montmorillonite particles, the results show that, due to its increasing concentration, the flocculation gel was fully adsorbed on the cuttings surface and occupied a large number of adsorption sites. At this point, additional flocculation gel molecules could not directly contact the cuttings, as the formerly adsorbed molecules hindered the adsorption of additional flocculation gel molecules on the cuttings surface. Therefore, the flocculation performance of a single flocculation gel molecule declined. Overall, the particle size and concentration of target rock cuttings should be thoroughly considered to determine the required concentration of the flocculation gel.

### 2.3. Effect of Flocculation Gel Concentration on Flocculation Performance

The working temperature of the flocculation gel is usually room temperature, but the effect of temperature on the flocculation gel performance should also be considered. Herein, interactions between the flocculation gel and solid-phase cuttings (represented by montmorillonite) at different temperatures were investigated, and the results, after statistical calculation, are shown in Figure 8. When the temperature was increased from 283 K to 313 K, the interaction energy between the flocculation gel and cuttings declined from −264 kcal/mol to −80 kcal/mol, which indicates that the interaction between the flocculation gel and cuttings becomes weaker with increasing temperature.

## 3. Conclusions

Oil-based drilling fluid is an oil–water emulsion system with oil as the external phase, which has the advantages of strong inhibition, high-temperature resistance, excellent lubricity, etc. It is broadly used in water-sensitive formations, deep formations with high temperatures and high pressures, as well as under other complex geological conditions. During an operation using oil-based drilling fluid, a lot of cuttings will enter the oil-based drilling fluid system due to the long, horizontal section, poor ability of the oil-based drilling fluid to carry cuttings, limited performance of solid-control equipment, and other reasons. This results in a high content of rock cuttings in the oil-based drilling fluid. These cuttings are hard to remove and eventually lead to a decline in the performance of the oil-based drilling fluid, hence further triggering serious incidents. Currently, introducing flocculation gel into oil-based drilling fluids is an effective approach to tackle the aforementioned issues. Herein, the formation of nano/micron-sized cuttings during the drilling operation and the selective flocculation of these particles by using a flocculation gel for oil-based drilling fluid was investigated via molecular simulation. The mechanism of the selective flocculation of the cuttings by the flocculation gel in the oil-based drilling fluid was analyzed, from which the following conclusions were obtained:The flocculation gel for oil-based drilling fluid selectively flocculates rock cuttings, and it is highly effective for flocculating electrically negative cuttings components, but it is ineffective for uncharged cuttings components.The flocculation gel for oil-based drilling fluid is ineffective for cuttings with strong hydrophilicity, while exerting almost no flocculation effect on cuttings with weak hydrophilicity.Within a limited concentration range, the flocculation effect becomes better with increasing flocculation gel concentrations, but the interaction between a single flocculation gel molecule and the cuttings will be diminished.The performance of the flocculation gel declines as the environmental temperature increases.

## 4. Methods

### 4.1. Modeling and Simulation Methods

Considering the mineral component profile of rock cuttings, a molecular model of the cuttings–flocculation gel system (represented by kaolinite and montmorillonite) employing oil-based drilling fluid was constructed to simulate and analyze the interactions between the flocculation gel for oil-based drilling fluid and clay minerals. Here, the crystal cell of kaolinite was adopted as the basic unit to construct the electrically neutral scaffolds of the clay mineral particles (with different hydrophilicity) in the cuttings. Montmorillonite was used as the basic unit to construct the electrically negative scaffolds of clay mineral particles in the cuttings. The constructed cationic molecular model is composed of a ternary polymer containing acrylamide, 2-acrylamide-2-methylpropanesulfonic acid, and dimethyl diallyl ammonium chloride. The molecular model of the system is illustrated in Figure 9; to simplify the research, the molecules of white oil in this system were replaced with C-13 alkane. The initial number of components used in the system was 60 kaolinite, 300 white oil, 1 flocculation gel, and 60 montmorillonite. In the present study, the simulation only focused on the flocculation effect of flocculation gel in paraffin-based oil.

### 4.2. Settings of Force Field and Cutoff Radius

During molecular dynamics simulations, the interactions between atoms are described by force fields. Force-field parameters are the basis of calculating the interactions between different atoms in molecular dynamics simulations. The model for the total potential energy is described using Equation (1):(1)$$E=E_{bond}+E_{angle}+E_{torsion}+E_{oop}+E_{cross}+E_{vdW}+E_{ele}$$
(2)$$E_{bond}=\sum_{bond}^{}K_{r}{\left(r-r_{0}\right)}^{2}$$
(3)$$E_{angle}=\sum_{angle}^{}k_{\theta }{\left(\theta -\theta_{0}\right)}^{2}$$
(4)$$E_{torsion}=\frac{V_{1}}{2}[1+\cos(\phi +f_{1})]+\frac{V_{2}}{2}[1-\cos(2\phi +f_{2})]+\frac{V_{3}}{2}[1+\cos(3\phi +f_{3})]$$
(5)$$E_{oop}=\sum_{i>j}^{n}f_{ij}(\frac{A_{ij}}{r_{ij}{}^{12}}-\frac{C_{ij}}{r_{ij}{}^{6}}+\frac{q_{i}q_{j}e^{2}}{4\pi \varepsilon_{0}r_{ij}})$$
where the bond contraction energy, bond angle-bending energy, dihedral angle torsion energy, nonplanar interaction energy (Equations (2)–(5)) and cross-term, represented by *E_bond_*, *E_angle_*, *E_trosion_*, *E_oop_*, and *E_cross_*, respectively, are bond potentials; the short-range van der Waals potential (*E_vdW_*) and long-range electrostatic potential (*E_ele_*) represent the potential energy for non-bond interactions.

In the simulation system, the non-bond interactions of all atoms are expressed by Equation (6) [14]:(6)$$U\left(r_{ab}\right)=U^{LJ}+U^{C}=4\varepsilon_{ab}\left[{\left(\frac{\sigma_{ab}}{r_{ab}}\right)}^{12}-{\left(\frac{\sigma_{ab}}{r_{ab}}\right)}^{6}\right]+\frac{q_{a}q_{b}}{4\pi \varepsilon_{0}r_{ab}}$$
where the parameters *q_a_* and *q_b_* are the charge quantities of atoms *a* and *b*, and *r_ab_* is the distance between the centers of atoms *a* and *b*. Parameter *ε*_0_ is the dielectric constant under vacuum, and *ε_ab_* controls the intensity of short-range interactions. The Lennard–Jones potential is a simple mathematical model used to compute the potential energy of interaction between two electrically neutral molecules or atoms. The Lennard–Jones potential parameters *σ_ab_* and *ε_ab_* are derived from the conventional Lorentz–Berthelot rule [20]:(7)$$\sigma_{ab}=\frac{\sigma_{a}+\sigma_{b}}{2}$$
(8)$$\varepsilon_{ab}=\sqrt{\varepsilon_{a}\varepsilon_{b}}$$

The long-range interactions are calculated by applying the PPPM summation algorithm with a convergence accuracy of 10^−8^ and a cutoff radius of 1.0 nm for non-bond interactions. The timestep is 1.0 fs. The periodic boundary condition (PPP) was used in this system, and the box sizes of the system along the x, y, and z directions were 4.63 nm, 4.46 nm, and 9.04 nm, respectively. For the accuracy of the calculated results, each simulation was repeated three times by varying the random number of simulations.

The simulation system included kaolinite, montmorillonite, white oil, and flocculation gel. In the system, kaolinite and montmorillonite molecules were described by the clay force field (ClayFF) [14], and the white oil and flocculation gel molecules were described by the CVFF force field [15]. The stage radius of the non-bond interactions was set to 1.0 nm. The equilibrium molecular dynamics simulation was run for 4.0 ns, and it was divided into two processes, during which the first 2.0 ns was used for system equilibrium and the last 2.0 ns was used to acquire data. An NVT ensemble was adopted during the simulation, and the system-temperature distribution was set to 283, 298, and 313 K for simulating the effect of the environmental temperature on the flocculation gel’s performance. The parameters used in this work are listed in Table 1.

### 4.3. Calculation Parameters

#### 4.3.1. Density Distribution Function

Analyzing the density distribution function of flocculation gel molecules in the trajectory of the molecular dynamics simulation serves as the basis for analyzing the localization of flocculation gel molecules on the surface of rock cuttings. The computer chunk/atom command in the LAMMPS software package was adopted for density calculations with real units in the simulation. The pore width was set to 60 Å, and the calculation-grid width of the one-dimensional and two-dimensional density distribution was set to 0.2 Å. The ordinate of the density distribution curve was represented by the mass density, and the starting point of the abscissa was the lower surface of the pore.

#### 4.3.2. Interaction Energy

By analyzing the variation in the interaction energy between the fluid and the wall, the state of the fluid on the interface can be determined. The interaction energy is the sum of hydrogen-bond energy, van der Waals energy, and electrostatic energy [19,21]. The stability of a surface adsorbate can be described by the interaction energy, for which a negative value indicates that the adsorbate can be adsorbed on the surface [22]. The interaction energy is calculated by applying Equation (9) [19,21,23,24]:(9)$$E_{interaction}=E_{AB}-(E_{A}+E_{B})$$
where *E_AB_* is the total energy of the A B complexes, and *E_A_* and *E_B_* are the energy of monomer A and monomer B, respectively.

#### 4.3.3. Mean Square Displacement and Diffusion Coefficient

The mean square displacement (*MSD*) is an important physical quantity for describing the characteristics of molecular dynamics and is a significant indicator for describing molecular motion. If molecular motion becomes more intense, the *MSD* will be higher, leading to the lower thermal stability of the substrate. The *MSD* is calculated using Equation (10):(10)$$MSD=\langle {\left|r_{i}\left(t\right)-r_{i}(0)\right|}^{2}\rangle$$
where *r_i_*(*t*) represents the position of molecule/atom *i* at time *t*, *r_i_*(0) represents the position of molecule/atom *i* at time = 0, and the angled brackets indicate the average value.

The slope of the time-dependence of the *MSD* is directly proportional to the diffusion coefficient of the atoms or molecules, i.e., 1/6 of the *MSD* slope equals the molecular diffusion coefficient. In three-dimensional space, if one direction is restricted, the diffusion coefficient then becomes 1/4 of the *MSD* slope. The diffusion coefficient can be solved based on the *MSD* by exploiting the Einstein relation [25]:(11)$$D=\frac{1}{4}\underset{t\to \infty }{\lim}\frac{\langle {\left|\vec{r}\left(t_{0}+t\right)-\vec{r}\left(t_{0}\right)\right|}^{2}\rangle }{t}$$

## Figures and Tables

**Figure 1 gels-08-00442-f001:**
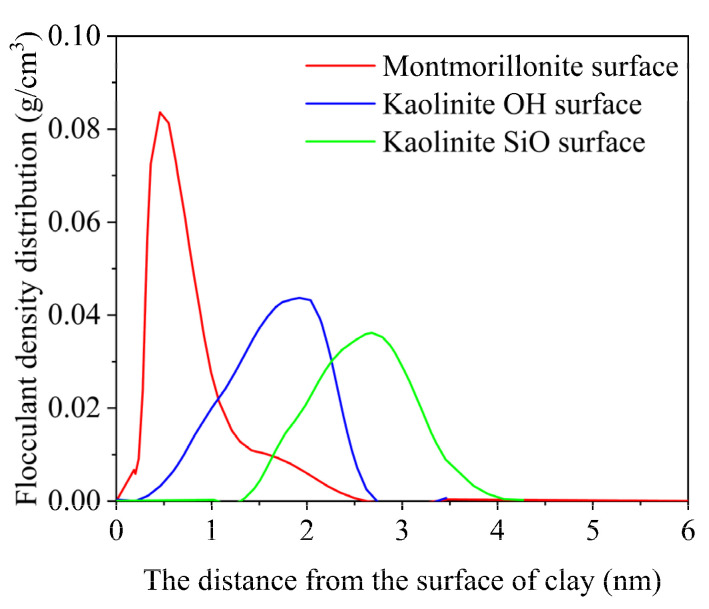
Density distribution of flocculation gel in the system.

**Figure 2 gels-08-00442-f002:**
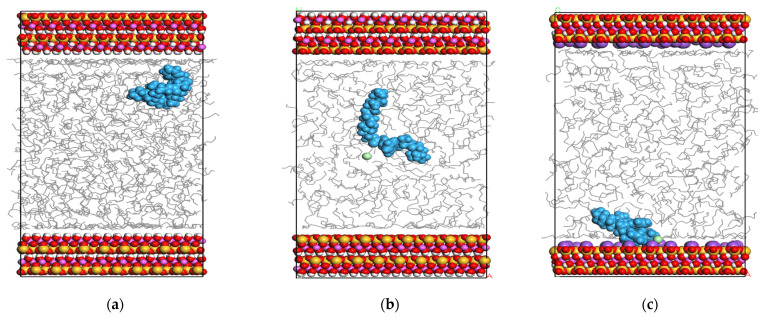
The distribution of particles in the system at equilibrium: (**a**) Particle distribution in the equilibrium of strongly hydrophilic surface cuttings–oil–soluble flocculant system, (**b**) Particle distribution in the equilibrium of weakly hydrophilic surface cuttings–oil–soluble flocculant system, (**c**) Particle distribution at equilibrium of charged cuttings–oil–soluble flocculant system. Blue molecules of bulk phase represent flocculation gel, gray molecules represent white oil, and green molecules represent the dissociated chloride ion.

**Figure 3 gels-08-00442-f003:**
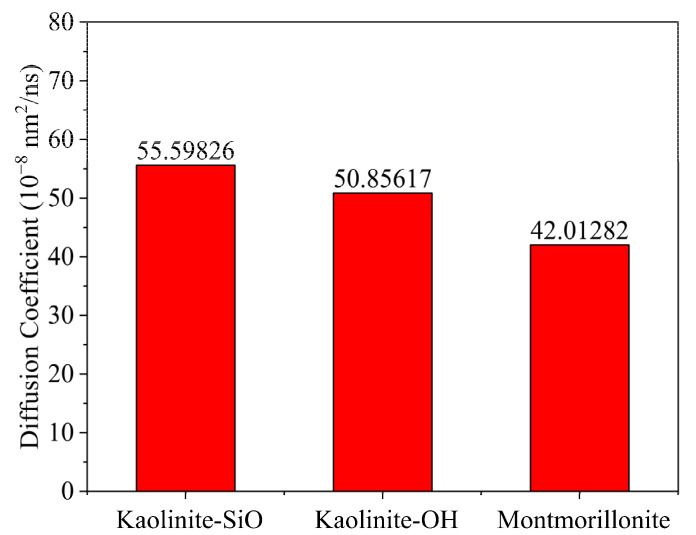
Variation of the flocculation gel diffusion coefficient after flocculating different clay minerals.

**Figure 4 gels-08-00442-f004:**
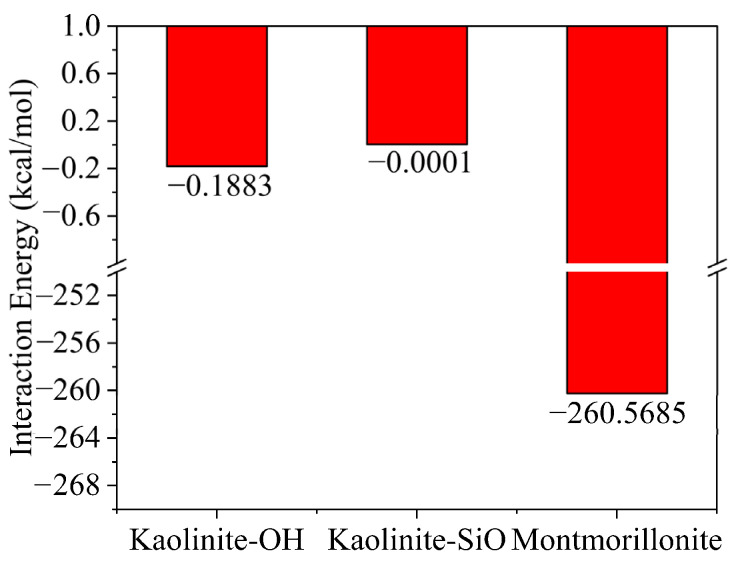
Correlation of interaction energy between flocculation gel and different clay minerals.

**Figure 5 gels-08-00442-f005:**
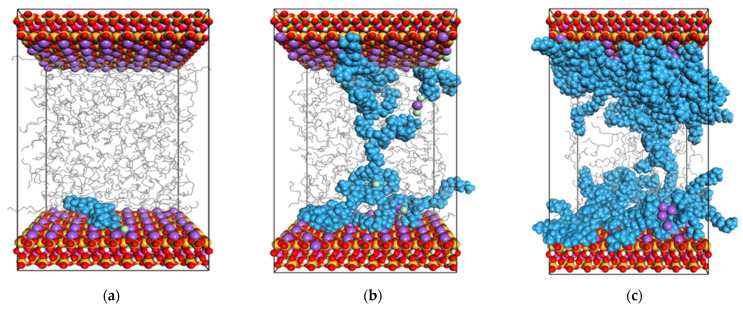
Molecular model of flocculation gel–cuttings systems with different flocculation gel concentrations: (**a**) Molecular distribution of flocculation gels with a number of 1, (**b**) molecular distribution of flocculation gels with a number of 10, (**c**) molecular distribution of flocculation gels with a number of 40.

**Figure 6 gels-08-00442-f006:**
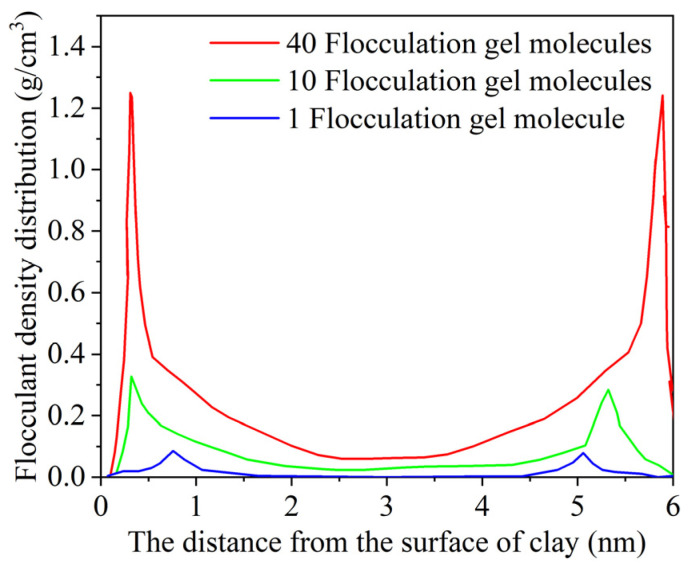
Density distribution in flocculation gel systems with different concentrations.

**Figure 7 gels-08-00442-f007:**
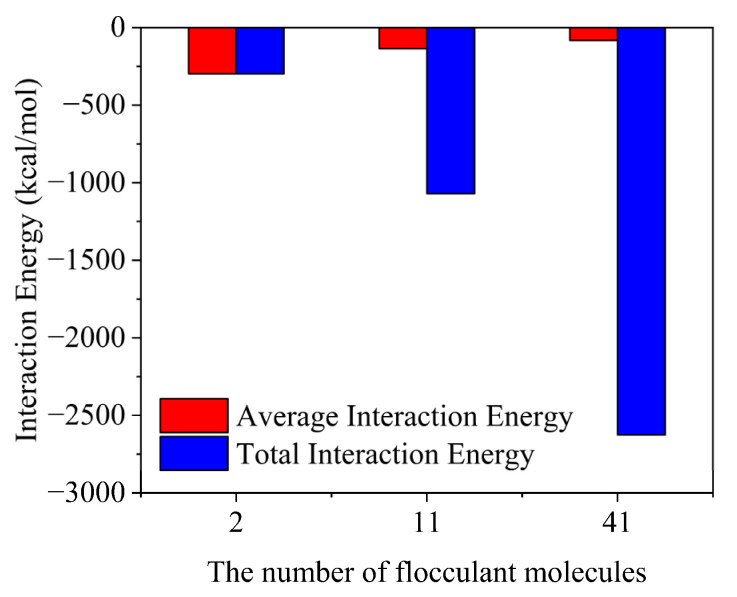
Variation of interaction energy in flocculation gel systems with different concentrations.

**Figure 8 gels-08-00442-f008:**
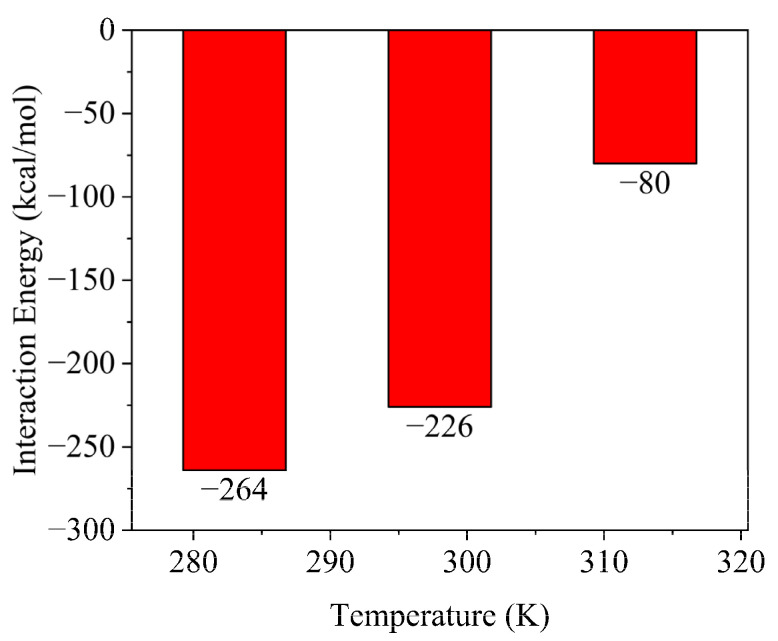
Temperature-dependent variation of interactions between flocculation gel and debris.

**Figure 9 gels-08-00442-f009:**
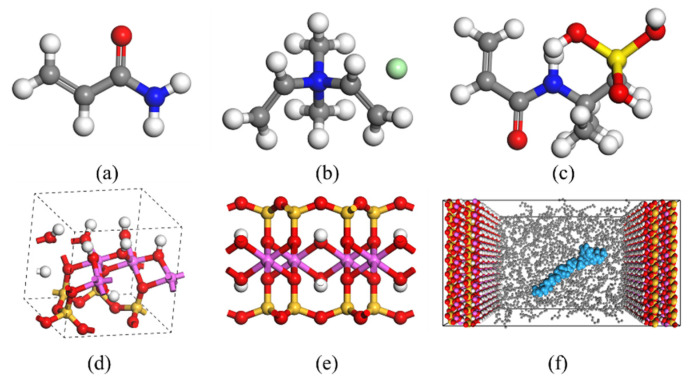
Molecular model of interactions between flocculation gel for oil-based drilling fluid and clay minerals: (**a**) acrylamide, (**b**) dimethyl diallyl ammonium chloride, (**c**) 2-acrylamide-2-methylpropanesulfonic acid, (**d**) kaolinite unit cell, (**e**) montmorillonite unit cell, and (**f**) surface cuttings-flocculation gel system.

**Table 1 gels-08-00442-t001:** Lennard–Jones parameters.

Atom	Mass (g/mol)	ε (kcal/mol)	σ (Å)
Montmorillonite			
O	15.999400	0.1554164124	3.1655200879
H	1.007970	0	0
Si	28.085500	0.0000018402	3.3019566252
Al	26.981540	0.0000013297	4.2713219316
Mg	24.305000	0.0000009030	5.2643258688
White oil			
C	12.011150	0.0389999952	3.8754094636
H	1.007970	0.0380000011	2.4499714540
Flocculation gel			
C	12.011150	0.0389999952	3.8754094636
O	15.999400	0.2280000124	2.8597848722
H	1.007970	0.0380000011	2.4499714540
N	14.006700	0.1669999743	3.5012320066
Cl	35.453000	0.1070000050	4.4462973121
Kaolinite			
Al	26.981540	0.0000013297	4.2713219316
Si	28.085500	0.0000018402	3.3019566252
O	15.999400	0.1554164124	3.1655200879
H	1.007970	0	0

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
