# Peer review of "Molecular Simulation of Interactions between High-Molecular-Polymer Flocculation Gel for Oil-Based Drilling Fluid and Clay Minerals"

_gels, 2022, doi:10.3390/gels8070442_

Round 1

Reviewer 1 Report

Molecular simulation of interactions between high molecular polymer flocculation gel for oil-based drilling fluid and clay minerals

Zhijun He et al.

The authors presented an interesting manuscript proposing a molecular simulation of flocculation gel for oil-based drilling fluid. This manuscript can be of great interest to the researchers and engineers that work in this area. In my opinion, the paper can be published in Gels after the authors improve the manuscript based on the major concerns stated below. The main point that could be improved is to further discuss the results and the methodology.

Major Concerns

1.       Although it is interesting to have a “direct” text showing an introduction, results, and conclusions, and just after the conclusions the authors present the methodology, in my opinion, the authors should warn the reader that the materials and methods section is placed after the conclusion. In the first time that one reads the manuscript, it is hard to understand the results and discussions, because the reader does not know which methodology was used to achieve those results.

2.       The authors talk about “oil-based drilling fluid”, but the reader does not know the composition of the analyzed oil-based drilling fluid. Is it an olefin or paraffinic base oil? Is it an inverted emulsion of oil in the water? Which are the other components?

3.       This simulation is valid for any oil-based drilling fluid? Which are the hypotheses used that must be further analyzed in future works (based on the idea of the real drilling fluid)?

4.       The authors should make clearer the methodology and discussions of the manuscript. Which part of the work is experimental and which part is simulation? After reading it more than three times the manuscript one can understand this point, but it must be clear for the reader the first time that he/she reads the paper.

5.       The authors say that “the molecules of white oil in this system are replaced by C-13 alkane”. One can understand that this simulation is valid only for paraffinic-based oil drilling fluid?

6.       If one has a real distribution of alkanes in the sample, the results would be different? Or assuming only C-13 alkane is a good approximation?

7.       The authors should make clearer which values they used to estimate the parameters presented in the model.

Reviewer 2 Report

The proposed article is quite interesting, good quality. But there are comments and recommendations.

1. It is necessary to add information about the composition of drilling fluid.

2. Enlarge section "Introduction". Add sources related to the research topic.

3. Section 4 "Materials and Methods" must be placed after the "Introduction".

4. "Conclusions" section should be at the end of article.

Author Response

Response to Reviewer 2 Comments

Point 1: It is necessary to add information about the composition of drilling fluid.

Response 1: Thank the reviewer very much. In this paper, the flocculation mechanism of single-agent flocculation gel in oil-based drilling fluid was studied by molecular simulation. The influence of other components of oil-based drilling fluid on the flocculation effect of flocculation gel has not been considered. The effects of other components of oil-based drilling fluids on their flocculation will be discussed in future work.

Point 2: Enlarge section "Introduction". Add sources related to the research topic.

Response 2: Thank the reviewer very much. Relevant references have been added to the introduction section after revision.

Specifically inserted/reworded:

“1.DONG Dazhong,ZOU Caineng,YANG Hua,WANG Yuman,LI Xinjing,CHEN Gengsheng. Progress and prospects of shale gas exploration and development in China[J]. ACTA PETROLEI SINICA, 2012, 33(S1): 107-114.” is cited in the Introduction section. (lines 390-391, page 13 in RM)

“2.Xingang Z, Jiaoli K, Bei L. Focus on the development of shale gas in China—Based on SWOT analysis[J]. Renewable and Sustainable Energy Reviews, 2013, 21: 603-613.” is cited in the Introduction section. (lines 392-393, page 13 in RM)

“4.Tonglou G U O. Key geological issues and main controls on accumulation and enrichment of Chinese shale gas[J]. Petroleum Exploration and Development, 2016, 43(3): 349-359.” is cited in the Introduction section. (lines 398-399, page 13 in RM)

“5.Yan L, Wang J, Zhang J, et al. Understanding and research on the drilling fluid technology for shale gas horizontal wells in south Sichuan, China[C]//International Petroleum Technology Conference. OnePetro, 2020.” is cited in the Introduction section. (lines 400-401, page 13 in RM)

Point 3: Section 4 "Materials and Methods" must be placed after the "Introduction".

Response 3: Thank the reviewer very much. After modification, Section 4 "Materials and methods" has been placed after "Introduction".

Point 4: "Conclusions" section should be at the end of article.

Response 4: Thank the reviewer very much. After modification, the "Conclusion" section is already at the end of the article.

Reviewer 3 Report

Review 

The authors carry out an equilibrium molecular dynamics simulation of interactions between flocculation gel and clay minerals. The results presented will be of use in both academia and industry. However, I believe the paper lacks lucidity and clarity in the presentation and is difficult to understand in its current form. The authors should proofread their manuscript keeping in mind that the results and details should be provided in such a way that the research can be reproduced by anyone else based solely on the details provided in the manuscript, and thus request the authors to rewrite the manuscript adding details and explanations of their analysis. I would recommend the paper for publication after the questions/concerns are properly addressed by the authors.

Comments:

  • The authors should provide the theory of the model and simulation details before the results and discussions section.

  • The authors should provide relevant citations explaining their statements on the hydrophilicity of the 3 systems (Line 94 - 103)

  • Figure 2 is difficult to follow without sufficient labels and detailed captions. The authors should explain in detail in the main text each of the figures referring specifically to the labels (a), (b), and (c). What is the green bead in figure2 b and c, and in figure 10b, and why is it non-bonded?

  • Section 4.1 materials: I am quite confused as to what and why the authors synthesized and where this experimental detail is used in this paper? This section seems out of place.

  • Section 4.2.1 - 4.2.3 can come before the results and discussion section and again, the authors need to write and explain in detail the figure 10 with reference to the proper labels. And also, expand on the figure caption, plus add labels to the figure.

In section 4.2.2, the expressions for the individual terms are missing. The parameter values used in each of the terms should also be mentioned.

  • In Figure 6 density distribution peaks near the walls. What kind of boundary condition was used in the system? What was the box size in real / simulation units? What are the finite size effects of the system? How many independent runs were carried out to get the results? And how long was the simulation required to run in terms of CPU hours i.e 4.0ns is equal to how many CPU hours? The authors should also test the dependence on the system size of the simulations. Also, the authors can make tables to provide the details of their simulation system: the number of each type of particles used in the simulation, the box size, the timestep used

  • Regarding the cutting of the surface, the authors mention the energy. The authors should add a paragraph explaining the dynamic bond formation in the simulations.  Are these bonds reversible or permanent? 

  • The references 1-2 suggested in the manuscript do not correlate with their sentences.  Reference 7 is missing. Line 68 -69 needs a reference.

  • What about the effects of gravity? Will it play a role as the flocculating system absorbs more clay components?

  • Please proofread for typos in spelling and formula,  and grammatical errors.

Author Response

Response to Reviewer 3 Comments

Point 1: The authors should provide the theory of the model and simulation details before the results and discussions section.

Response 1: Thank the reviewer very much. After modification, Section 4 "Materials and methods" has been placed after "Introduction" and before the results and discussions section, the "Conclusion" section is already at the end of the article.

Point 2: The authors should provide relevant citations explaining their statements on the hydrophilicity of the 3 systems (Line 94 - 103)

Response 2: Thank the reviewer very much. In this study, montmorillonite was used as an electron-negative scaffold for constructing clay mineral particles in cuttings. The influence of its chargeability on the flocculation effect of the flocculation gel was mainly considered, and the influence of its hydrophilicity was not considered. This will continue to be studied in the next work. The hydroxyl surface and the silica surface of kaolinite represent the strong hydrophilic surface and the weak hydrophilic surface, respectively. The influence of the hydrophilicity of cuttings on the flocculation effect of flocculation gel can be obtained by studying the two surfaces.

Specifically inserted/reworded:

“The overall characteristics of the surface of kaolinite lingstone are hydrophilic. There are multiple relatively strong hydrogen bonds and surface hydroxyl groups on the octahedral surface. The contact angle of the surface is equal to zero, so the surface has strong hydrophilicity. The tetrahedron has numerous siloxy groups, and the water molecules on the surface are kept together, maintaining the deformed shape of the droplet, indicating that the surface of the tetrahedral kaolinite is hydrophobic.” is added. (lines 185-191, page 6 in RM)

“22.Šolc R, Gerzabek M H, Lischka H, et al. Wettability of kaolinite (001) surfaces—molecular dynamic study[J]. Geoderma, 2011, 169: 47-54.” is cited in the section. (lines 440-441, page 13 in RM)

Point 3: Figure 2 is difficult to follow without sufficient labels and detailed captions. The authors should explain in detail in the main text each of the figures referring specifically to the labels (a), (b), and (c). What is the green bead in figure2 b and c, and in figure 10b, and why is it non-bonded?

Response 3: Thank the reviewer very much. The green beads represent dissociated chloride ions and are therefore non-binding.

Specifically inserted/reworded:

“Figure 4. Schematics of particle distribution in the system at equilibrium; : (a) Schematic diagram of particle distribution in the equilibrium of strongly hydrophilic surface cuttings-oil-soluble flocculant system, (b) Schematic diagram of particle distribution in equilibrium of weakly hydrophilic surface cuttings-oil-soluble flocculant system, (c) Schematic diagram of particle distribution at equilibrium of charged cuttings-oil-soluble flocculant system, blue molecules of bulk phase represent flocculation gel and gray molecules represent white oil, green molecule represents the dissociated chloride ion. “(lines 200-206, page 6 in RM)

Point 4: Section 4.1 materials: I am quite confused as to what and why the authors synthesized and where this experimental detail is used in this paper? This section seems out of place.  

Response 4: Thank the reviewer very much. After modification, Section 4 "Materials and methods" has been placed after "Introduction" and before the results and discussions section, the "Conclusion" section is already at the end of the article.

Point 5: Section 4.2.1 - 4.2.3 can come before the results and discussion section and again, the authors need to write and explain in detail the figure 10 with reference to the proper labels. And also, expand on the figure caption, plus add labels to the figure.

In section 4.2.2, the expressions for the individual terms are missing. The parameter values used in each of the terms should also be mentioned.

Response 5: Thank the reviewer very much. After modification, Section 4 "Materials and methods" has been placed after "Introduction" and before the results and discussions section, the "Conclusion" section is already at the end of the article.

Specifically inserted/reworded:

“Figure 2. Molecular model of interactions between flocculation gel for oil-based drilling fluid and clay minerals.: (a) acrylamide, (b) dimethyl diallyl ammonium chloride, (c) 2-acrylamide-2-methylpropanesulfonic acid, (d) Kaolinite unit cell, (e) Montmorillonite unit cell, (f) Surface cuttings-flocculation gel system” (lines 110-113, page 3 in RM)

“where, the bond contraction energy, bond angle bending energy, dihedral angle torsion energy, nonplanar interaction energy and cross-term, represented by Ebond, Eangle, Etrosion, Eoop, and Ecross, respectively, are bond potentials; the short-range van der Waals potential (EvdW) and long-range electrostatic potential (Eele) represent the potential en-ergy for non-bond interactions.” (lines 119-123, page 4 in RM)

Point 6: In Figure 6 density distribution peaks near the walls. What kind of boundary condition was used in the system? What was the box size in real / simulation units? What are the finite size effects of the system? How many independent runs were carried out to get the results? And how long was the simulation required to run in terms of CPU hours i.e 4.0ns is equal to how many CPU hours? The authors should also test the dependence on the system size of the simulations. Also, the authors can make tables to provide the details of their simulation system: the number of each type of particles used in the simulation, the box size, the timestep used

Response 6: Thank the reviewer very much.

Specifically inserted/reworded:

“The time step is 1.0 fs. The periodic boundary conditions (PPP) was used in this system, and the box sizes of the system along the x, y, and z directions are 4.63 nm, 4.46 nm, and 9.04 nm, respectively. For the accuracy of the calculated results, each simulation was repeated 3 times by varying the random number of simulations.” (lines 133-136, page 4-5 in RM)

Especially, in order to reduce calculation errors due to the small size, the length of the model in the z-direction was increased to be much larger than the cut-off distance. For each simulation, 28 hours was spent to calculate by using CPU with 56 core. As the interaction between flocculant and clay is mainly studied, the model size is mainly determined by the size of the flocculant, which is related to the molecular weight of the flocculant. It will be further explored in our future work.

Point 7: Regarding the cutting of the surface, the authors mention the energy. The authors should add a paragraph explaining the dynamic bond formation in the simulations.  Are these bonds reversible or permanent?

Response 7: Thank the reviewer very much. The flocculation process of cationic flocculants is a physical process, which do not involve the formation and destruction of chemical bonds.

Point 8: The references 1-2 suggested in the manuscript do not correlate with their sentences.  Reference 7 is missing. Line 68 -69 needs a reference.

Response 8: Thank the reviewer very much. References 1-2 have been revised as relevant references, meanwhile, reference 7(It is now reference 10) has been cited in the manuscript and relevant references have been added on lines 68-69.

Specifically inserted/reworded:

“1.          DONG Dazhong,ZOU Caineng,YANG Hua,WANG Yuman,LI Xinjing,CHEN Gengsheng. Progress and prospects of shale gas exploration and development in China[J]. ACTA PETROLEI SINICA, 2012, 33(S1): 107-114.” (lines 403-404, page 13 in RM)

“5.          Yan L, Wang J, Zhang J, et al. Understanding and research on the drilling fluid technology for shale gas horizontal wells in south Sichuan, China[C]//International Petroleum Technology Conference. OnePetro, 2020.” (lines 415-416, page 13 in RM)

“Exploiting molecular simulation methods in developing flocculation gel can provide guidance for understanding the mechanism of selective flocculation on the micro-scale[10,12–14].” (lines 415-416, page 13 in RM)

“24.        Vajihinejad V, Gumfekar S P, Bazoubandi B, et al. Water soluble polymer flocculants: synthesis, characterization, and per-formance assessment[J]. Macromolecular Materials and Engineering, 2019, 304(2): 1800526.” (lines 453-454, page 14 in RM)

Point 9: What about the effects of gravity? Will it play a role as the flocculating system absorbs more clay components?

Response 9: Thank the reviewer very much. Without considering the influence of gravity, this paper only studies the ability of flocculants to adsorb cuttings, and will discuss the adsorption of flocculants under the action of more cuttings in the next step.

Point 10: Please proofread for typos in spelling and formula, and grammatical errors.

Response 10: Thank the reviewer very much.

Specifically inserted/reworded:

“where, the bond contraction energy, bond angle bending energy, dihedral angle torsion energy, nonplanar interaction energy and cross-term, represented by Ebond, Eangle, Etrosion, Eoop, and Ecross, respectively, are bond potentials; the short-range van der Waals potential (EvdW) and long-range electrostatic potential (Eele) represent the potential energy for non-bond interactions.” (lines 119-125, page 4 in RM)

“Meanwhile, the flocculation gel molecules should be chain molecules with a small number of branches or cross-linked structures, and should have a relatively large molecular weight, generally around 106, to form bridging structures between different suspended particles[24].” (lines 67-70, page 4 in RM)

Round 2

Reviewer 1 Report

Molecular simulation of interactions between high molecular polymer flocculation gel for oil-based drilling fluid and clay minerals

Zhijun He et al.

The authors answered all my concerns and improved the manuscript. In my opinion, the paper can be published in Gels after the authors improve the manuscript based on the minor concerns stated below.

Minor Concerns

1.       Table 1 was inserted in the revised manuscript, but it was not cited and discussed in the text.

2.       The legend of Table 1 is “Lennard-Jones parameters”, but the authors did not present the “Lennard-Jones parameters” before. One cannot understand if “Lennard-Jones” is a model, equation, or empiric parameter. The authors must further describe this point.

3.       In my opinion, the authors could be clearer about the “experimental” results and the “numerical” results. It is not clear to the reader which contribution of the manuscript is experimental and which contribution is based on the mathematical model. Ok, the authors describe in section 2.1 “Synthesis of flocculated gel materials”. But, after that, which information is experimental? It seems to the reader that all the results are from the model. The authors could be more specific in showing which contribution is experimental and which is numerical.  

Author Response

Response to Reviewer 1 Comments

Point 1: Table 1 was inserted in the revised manuscript, but it was not cited and discussed in the text.

Response 1: Thank the reviewer very much. We have corrected it.

Specifically inserted/reworded:

“The parameters used in this work were listed in Table 1.” is added. (lines 148-149, page 5 in RM)

Point 2: The legend of Table 1 is “Lennard-Jones parameters”, but the authors did not present the “Lennard-Jones parameters” before. One cannot understand if “Lennard-Jones” is a model, equation, or empiric parameter. The authors must further describe this point.

Response 2: Thank the reviewer very much. The Lennard-Jones potential is a simple mathematical model used to compute the potential energy of interaction between two electrically neutral molecules or atoms. We have cited it in RM.

Specifically inserted/reworded:

“The Lennard-Jones potential is a simple mathematical model used to compute the potential energy of interaction between two electrically neutral molecules or atoms.”is added. (lines 130-132, page 5 in RM)

“25. S. Xiao, S.A. Edwards, F. Gräter, A New Transferable Forcefield for Simulating the Mechanics of CaCO3 Crystals, The Journal of Physical Chemistry C 115 (2011) 20067-20075.” is cited in the section.

(lines 390-391, page 13 in RM)

Point 3: In my opinion, the authors could be clearer about the “experimental” results and the “numerical” results. It is not clear to the reader which contribution of the manuscript is experimental and which contribution is based on the mathematical model. Ok, the authors describe in section 2.1 “Synthesis of flocculated gel materials”. But, after that, which information is experimental? It seems to the reader that all the results are from the model. The authors could be more specific in showing which contribution is experimental and which is numerical.

Response 3: Thank the reviewer very much. The suggestion has been taken. The lonely results experimental has been deleted. ( lines 84-92, page 2-3 in RM)

Reviewer 3 Report

I thank the authors for their revision of the manuscript. However, some points are still not clear to me which are mentioned below. I believe once the authors respond to the below-mentioned comments, the article can be recommended for publication in the journal.

  • Response 3: It is not clear why the captions of figures 4 are labeled as schematics. Are they not real simulation data snapshots produced using some data visualization software?

  • Response 4: It is still not very clear. Why is section 2.1 presented in this manuscript? The authors should make it clear why they have the experimental synthesis section in the modeling paper?

  • Response 5: The equations are still not provided for some of the following: Ebond, Eangle, Etrosion, Eoop, and Ecross etc. 

  • Response 6: 

    • “The simulation system includes kaolinite, montmorillonite, white oil, and flocculation gel.” The authors should mention the number of each of the components of the system used in the simulations.

    • “The periodic boundary conditions (PPP) was used in this system”. From figures 4,7 and 8, it seems that a fixed boundary was imposed along the “Z” direction. The X-axis of Figure 8 is labeled as “Distance from wall (nm)”.  In Fig. 7 and 8, why the boundary layers are not moving inward from the top & bottom faces of the simulation box? If it is a PPP simulation, the wall layers can very well be placed in the middle of the box with the surrounding white oil and the flocculating gel system surrounding it from all sides. The authors are requested to produce a different simulation snapshot when the layers are in the middle and the flocculating gel seems to cover from top to bottom, or else mention explicitly that the figures are chosen for visual clarity but indeed the simulations are PPP and not a PPF. Also, in Fig. 8 clarify what the authors meant by distance from wall.

    • Fig. 7 caption should be more elaborate.
  • Response 10:  Please proofread for typos in spelling and formula, and grammatical errors.

    • See line 46 “manuscript”  with a mismatched font . 

    • The references have “[J]” in them.
